# Introduce Ce^3+^ Ions to Realize Enhancement of C+L Band Luminescence of KMnF_3_: Yb, Er Nanoparticles

**DOI:** 10.3390/nano13152153

**Published:** 2023-07-25

**Authors:** Hao Cui, Daguang Li, Yu Yang, Yuewu Fu, Yanhui Dong, Jing Yin, Weiping Qin, Zhixu Jia, Dan Zhao

**Affiliations:** State Key Laboratory on Integrated Optoelectronics, College of Electronic Science & Engineering, Jilin University, Changchun 130012, Chinawpqin@jlu.edu.cn (W.Q.); jiazx@jlu.edu.cn (Z.J.)

**Keywords:** KMnF_3_, nanoparticles, active-shell, core-shell, 1.5 μm

## Abstract

Polymer-based waveguide amplifiers are essential components in integrated optical systems, as their gain bandwidths directly determine the operating wavelength of optical circuits. However, development of the wideband gain media has been challenging, making it difficult to fabricate devices with broadband amplification capability. Rare earth ion-doped nanoparticles (NPs) are a key component in the gain media, and their full width at half maximum (FWHM) of the emission peak decides the final gain bandwidth of the gain media. Here, KMnF_3_: Yb, Er, Ce@KMnF_3_: Yb NPs with the broad full width at half maximum (FWHM) of the emission peak covering the S+C band was prepared. The NPs were synthesized using a hydrothermal method, and the FWHM of the emission peak of NPs reached 76 nm under the excitation of a 980 nm laser. The introduction of Ce^3+^ ions and a core-shell structure coating greatly enhanced the emission intensity of NPs at C band. Since KMnF_3_: Yb, Er, Ce@KMnF_3_: Yb NPs have exceptional broadband luminescence properties at C band, KMnF_3_: Yb, Er, Ce@KMnF_3_: Yb NPs can be the potential gain medium in the future polymer-based waveguide amplifiers.

## 1. Introduction

Rare earth ion-doped nanoparticles (NPs) have attracted much attention since their excellent optical properties. The fluoride nanoparticles can be used in solar cells, biomarkers, in vivo drug delivery, photocatalysis, biological imaging, cancer therapy, optical waveguide amplifiers, anti-counterfeiting, and other fields [1,2,3,4,5,6,7,8,9,10,11,12]. Rare earth ion-doped nanoparticles (NPs) have attracted much attention due to their excellent optical properties compared with the conventional metal nanoparticles [13,14,15]. Recently, the rare earth ions-doped NPs have gained widespread acceptance as a key component in the gain medium for the fabrication of waveguide amplifiers [16,17]. The gain bandwidth of a waveguide amplifier was determined by the emission bandwidth of the NPs doped in the gain medium. Due to the ^4^I_13/2_ → ^4^I_15/2_ transition of Er^3+^ ions located at 1.53 μm, the luminescence peaks of Er^3+^ ions coincide well with the C band [18]. Therefore, Er^3+^ ions-doped NPs can be utilized in C-band waveguide amplifiers. Specially, the waveguide amplifiers based on KMnF_3_: Yb, Er NPs have been reported recently because of the broadband luminescence properties in C band of the NPs [19]. However, the low luminescence intensity of KMnF_3_: Yb, Er NPs makes the NPs difficult for practical requirements. In order to obtain Er^3+^-doped optical waveguide amplifier with high-gain, it is necessary to obtain a gain medium with intensive luminescence characteristics. The improvement of the luminescence properties of KMnF_3_: Yb, Er NPs becomes a challenge. So far, several methods have been proposed to enhance the luminescence intensity of Er^3+^-doped fluoride at 1.53 μm. The first strategy is to increase the transition rate from the highly excited state energy level of Er^3+^ to the ^4^I_13/2_ energy level [20,21], thereby increasing the population of Er^3+^ at the ^4^I_13/2_ energy level. And finally, the luminescence intensity at 1.53 μm can be enhanced due to the increase of radiative transition ^4^I_13/2_ → ^4^I_15/2_ energy level. The second strategy is to coat the surface of nanometer-sized core particles with a shell, which can repair the surface defect of the cores and effectively shield the emissive lanthanide ions near the surface from quenchers in the surroundings. The core-shell structure enhances the luminescence intensity at 1.53 μm of Er^3+^, thereby improving the luminescence properties of KMnF_3_: Yb, Er NPs [22,23,24,25,26].

In this paper, we choose KMnF_3_ as the matrix material since the FWHM of emission peak of the Er^3+^ ions-doped KMnF_3_ NPs is broader than that of the typical NaYF_4_ [27]. We prepared KMnF_3_: Yb, Er NPs via a hydrothermal method, and the influence of Yb^3+^ and Er^3+^ concentrations on the luminescence intensity at 1.53 μm of the NPs was investigated. And the doping concentrations of Yb^3+^ and Er^3+^ ions were found when Er^3+^ ion had the strongest luminescence at 1.5 μm. In order to further enhance the luminescence intensity at 1.53 μm, Ce^3+^ ions were introduced due to the existence of the energy transfer process between Ce^3+^ and Er^3+^: ^4^I_11/2_ (Er^3+^) + ^2^F_5/2_ (Ce^3+^) → ^4^I_13/2_ (Er^3+^) + ^2^F_7/2_ (Ce^3+^) [28]. We also investigated the relationship between the Ce^3+^ concentration and the luminescence intensity at 1.53 μm of KMnF_3_: Yb, Er, Ce NPs. And the doping concentration of Ce^3+^ ion was found when Er^3+^ ion had the strongest luminescence at 1.5 μm. In order to reduce surface quenching, a core-shell structure was also introduced. Finally, we obtained core-shell NPs (KMnF_3_: Yb, Er, Ce@KMnF_3_: Yb) with strong luminescence at 1.53 μm.

## 2. Experiment Details

YCl_3_·6H_2_O (99.99%), YbCl_3_·6H_2_O (99.99%), ErCl_3_·6H_2_O (99.99%), CeCl_3_·6H_2_O(99.99%) were purchased from Yutai Chemical Reagent, Shandong, China. Oleic Acid (OA, 500 mL) was purchased from Alpha Aesop Company, Shanghai, China. Ethanol (98%) and cyclohexane (99%) was obtained from Beijing Fine Chemical Company, Beijing, China. KOH (98%), KF·2H_2_O (98%), MnCl_2_·2H_2_O (98%) were purchased from Aladdin, Shanghai, China. All chemicals were used without further purification. 

### 2.1. Preparation of KMnF_3_: Yb, Er, Ce NPs

First, 12 mmol KOH was dropped in a 50 mL beaker containing 10 mL oleic acid, 5 mL ethanol and 3 mL deionized water under stirring. About 30 min later, another water solution of 0.4 mmol chloride (the four chlorides of YCl_3_·6H_2_O, YbCl_3_·6H_2_O, ErCl_3_·6H_2_O, CeCl_3_·6H_2_O, and MnCl_2_·2H_2_O are 0.4 mmol in total) was also added drop by drop. About 30 min later, a 1 mL water solution of 3.5 mmol KF·2H_2_O was added dropwise. After stirring thoroughly, the solution was transferred to a hydrothermal reactor and heated to 200 °C for 3 h. The reaction solution was naturally cooled to room temperature, and the obtained KMnF_3_: Yb, Er, Ce nanoparticles were washed three times with ethanol and cyclohexane and dried to obtain powdery KMnF_3_: Yb, Er, Ce nanoparticles. 

Next, 0.2 mmol of the as-prepared KMnF_3_: Yb, Er, Ce nanoparticles were dissolved in cyclohexane as core to induce the subsequent shell.

### 2.2. Preparation of KMnF_3_: Yb, Er, Ce@KMnF_3_: Yb Core-Shell NPs

Next, 12 mmol KOH was dropped in a 50 mL beaker containing 10 mL oleic acid, 5 mL ethanol, and 3 mL deionized water under stirring. About 30 min later, another water solution of 0.4 mmol chloride (the four chlorides of YCl_3_·6H_2_O, YbCl_3_·6H_2_O, ErCl_3_·6H_2_O, CeCl_3_·6H_2_O, and MnCl_2_·2H_2_O are 0.4 mmol in total) was also added drop by drop. About 30 min later, a 1 mL water solution of 3.5 mmol KF·2H_2_O was added dropwise. At last, a 5 mL solution containing core NPs was dropped into the beaker. After stirring thoroughly, the solution was transferred to a hydrothermal reactor and heated to 200 °C for 12 h. The reaction solution was naturally cooled to room temperature, and the obtained KMnF_3_: Yb, Er, Ce@KMnF_3_: Yb nanoparticles were washed three times with ethanol and cyclohexane and then dried to obtain powdery KMnF_3_: Yb, Er, Ce@KMnF_3_: Yb nanoparticles. 

### 2.3. Characterization

The phase of the NPs was characterized by X-ray powder diffraction (XRD) (Model Rigaku Ru-200b), using a nickel-filtered Cu-Kα radiation (λ = 1.5406 Å), and the scan ranged from 10° to 70°. The morphology of the particles was characterized by JEM-2100F electron microscope (Tokyo, Japan) at 200 KV. Under the excitation of a 980 nm laser diode, the emission spectrum at 1.53 μm of the sample was collected with a SPEX1000M spectrometer (HORIBA Group, Kyoto, Japan) at room temperature, and the fixed power density was 70 W·cm^2^ (the slit width was 0.2 mm). 

## 3. Results and Discussion

### 3.1. Crystal Structure and Morphology

The KMnF_3_: Yb, Er and NaYF_4_: Yb, Er NPs were prepared, and their normalized up-conversion luminescence spectrum and spectrum at 1.53 μm excited by 980 nm laser were shown in Figure 1. It can be seen the luminescence intensity at the 522 nm (^2^H_11/2_ → ^4^I_15/2_) and 540 nm (^4^S_3/2_ → ^4^I_15/2_) of Er^3+^ was reduced due to the energy transfer between Mn^2+^ and Er^3+^ in the KMnF_3_: Yb, Er NPs. Therefore, the green up-conversion luminescence of KMnF_3_: Yb, Er NPs was weakened, and the red up-conversion luminescence was highlighted. Thus, KMnF_3_: Yb, Er NPs exhibits red up-conversion luminescence at 980 nm excitation. It can be seen in the spectrum of Figure 1b that KMnF_3_: Yb, Er NPs has two luminescence peaks at 1490 nm and 1530 nm, and its FWHM is about 10 nm wider than that of NaYF_4_: Yb, Er NPs. We believe that difference of emission spectrum at 1.53 μm was due to the different symmetry of the crystal fields of the two materials.

In order to enhance the luminescence intensity at 1.53 μm of KMnF_3_: Yb, Er NPs, we introduced Ce^3+^ ions and prepared an active core-shell structure. The phase structure of the resulting product was analyzed on a Model Rigaku Ru-200b X-ray powder diffractometer (XRD) with nickel-filtered Cu K_α_ radiation (λ = 1.5406 Å). We successfully synthesized KMnF_3_: 18Yb, 2Er, KMnF_3_: 18Yb, 2Er, 4Ce and KMnF_3_: 18Yb, 2Er, 4Ce@KMnF_3_: 20Yb NPs by hydrothermal method. The diffraction peak was good agreement with the literature value (JCPDS: 82-1334). The diffraction peaks at 21.2°, 30.1°, 37.2°, 43.1°, 48.48°, 53.5°, 62.5° can be ascribed to the (100), (110), (111), (200), (210), (211), and (220) planes. The diffraction peaks of the samples can be indexed to the cubic phase KMnF_3_. The X-ray diffraction pattern of the sample is shown in Figure 2. In KMnF_3_ crystals, Mn^2+^ ions have an ionic radius of 80 Å. The ionic radii of doped ions are as follows: Er^3+^ (88.1 Å), Yb^3+^ (85.8 Å), Ce^3+^ (103.4 Å). The ionic radius of the Yb^3+^ ion and the Er^3+^ ion is similar to that of the Mn^2+^ ion, so the Yb^3+^ ion and Er^3+^ ion can occupy the lattice site of the Mn^2+^ ion well. The ionic radius of the Ce^3+^ ion is slightly larger than that of the Mn ion, but the doped Ce^3+^ ion has little effect on the crystal phase of the KMnF_3_ crystal due to the low doping concentration of the Ce^3+^ ion.

The schematic diagram and TEM of KMnF_3_:18Yb, 2Er, 4Ce@KMnF_3_: 20Yb core-shell structured nanoparticles were shown in the Figure 3 and Figure 4. The cyan pellets represent the core nanoparticles, and the red pellets represent the shell material.

### 3.2. Optical Properties

#### 3.2.1. Effect of the Concentration of Yb^3+^ and Er^3+^ on the Luminescence Properties of KMnF_3_: Yb, Er NPs

Most materials exhibit Stokes-shifted, also known as down-conversion emission, where each emitted photon has lower energy than the absorbed photon. However, there are also materials that have the ability to generate anti-Stokes shift luminescence, where the emitted photons have higher energy than the photons used for excitation. Two-photon absorption-based luminescence and second harmonic generation are two examples of anti-Stokes processes that require high-energy pulsed lasers as excitation sources. Depending on the lifetime of the excited state, the two-photon or multi-photon processes require simultaneous or nearly simultaneous absorption of two coherent near-infrared (NIR) photons at high excitation power densities (~10^6^ W cm^−2^), due to the small two-photon absorption cross-section. From the research on macroscopic inorganic crystals, three major up-conversion mechanisms have been elucidated: (i) ground-state absorption combined with excited-state absorption (GSA/ESA); (ii) energy transfer up-conversion (ETU); and (iii) photon avalanche. Among these categories, ETU is considered to be the most effective up-conversion (UC) mechanism. When a macroscopic crystal is simply doped with a low concentration of a trivalent rare-earth (RE) element, the interactions between ions can be neglected, and GSA/ESA is responsible for the UC process. As the doping concentration increases, the interactions between ions become significant, and the probability of energy transfer between ions in the excited state and the ETU mechanism increases. One approach to improve up-conversion (UC) efficiency is to use sensitizers with a simple energy scheme and high absorption cross-section in the near-infrared (NIR) region. These sensitizers absorb photon energy and transfer it to the up-conversion activators. As we all know, Yb^3+^ was often used as a sensitizer due its large absorption cross section at 980 nm in the Er^3+^ and Yb^3+^ co-doped system, due to the large energy overlap between the ^2^F_5/2_ → ^2^F_7/2_ energy level transition of Yb^3+^ and the transition of many energy levels of Er^3+^. Therefore, Yb^3+^ continuously absorbs 980 nm photons and transfers energy to Er^3+^ to populate the high-energy level of Er^3+^ under the excitation of a 980 nm laser diode. And the high-energy level of Er^3+^ was radiated to the low-energy level through transition, thereby emitting up-conversion luminescence and luminescence at 1.53 μm. In the KMnF_3_: Yb, Er NPs, the Mn^2+^ will undergo four energy transfer processes with Er^3+^: ^6^A_1_ (Mn^2+^) + ^2^H_9/2_ (Er^3+^) → ^4^T_1_ (Mn^2+^) + ^4^I_13/2_ (Er^3+^); ^6^A_1_ (Mn^2+^) + ^2^H_11_ (Er^3+^) → ^4^T_1_ (Mn^2+^) + ^4^I_13/2_ (Er^3+^); ^6^A_1_ (Mn^2+^) + ^4^S_3/2_ (Er^3+^) → ^4^T_1_ (Mn^2+^) + ^4^I_13/2_ (Er^3+^); ^4^T_1_ (Mn^2+^) + ^4^I_15/2_ (Er^3+^) → ^6^A_1_ (Mn^2+^) + ^4^F_9/2_ (Er^3+^) [29]. The energy transfer process in KMnF_3_: Yb, Er NPs was shown in Figure 5. 

The concentration of Yb^3+^ and Er^3+^ could greatly affect the luminescence intensity of the up-conversion luminescence and luminescence at 1.53 μm of the KMnF_3_: Yb, Er NPs. In order to explore the best concentration of Yb^3+^ and Er^3+^ in KMnF_3_: Yb, Er NPs, we performed a series of experiments. First, the concentration of Er^3+^ was doped at 1 mmol% and the concentration of Yb^3+^ (12%, 14%, 16%, 18%, 20%, 22%) was changed, then their up-conversion luminescence emission spectrum and luminescence at 1.53 μm emission spectrum was tested under the excitation of a 980 nm laser diode. Figure 6 shows the up-conversion luminescence emission spectrum and luminescence at 1.53 μm emission spectrum with varying Yb^3+^ concentration under the excitation of a 980 nm laser. We found that the intensity of the luminescence at 1.53 μm gradually increased with increasing Yb^3+^ concentration from 12% to 18% (as shown in Figure 6b). This was due to the fact that increasing the concentration of Yb^3+^ doping can enhance the absorption of pump energy. However, it is important to note that at higher concentrations of Yb^3+^ ions (>18%), the luminescence intensity at 1.53 μm starts to decrease due to concentration quenching caused by the high Yb^3+^ concentration. Second, the best concentration of Er^3+^ was determined. The concentration of Yb^3+^ was doped at 18 mmol% and the Er^3+^ concentration (0.5%, 1%, 2%, 3%, 4%) was changed, then their up-conversion luminescence emission spectrum and luminescence at 1.53 μm emission spectrum was tested under the excitation of a 980 nm laser diode. Figure 7 shows the up-conversion luminescence emission spectrum and luminescence at 1.53 μm emission spectrum with varying Er^3+^ concentration under the excitation of a 980 nm laser. We found that the intensity of the luminescence at 1.53 μm gradually increased with increasing Er^3+^ concentration from 0.5% to 2% (as shown in Figure 7b). This was due to the fact that the Er^3+^ ion is the luminescence center ion, and the increase of luminescence center increases the luminescence intensity. However, it is important to note that at higher concentrations of Er^3+^ ions (>2%), the luminescence intensity at 1.53 μm starts to decrease due to concentration quenching caused by the high Er^3+^ concentration.

Through the above experiments, we determined that the best doping concentration of Yb^3+^ and Er^3+^ in the KMnF_3_ matrix were 18% and 2%. 

#### 3.2.2. Effect of Ce^3+^ Concentration on the Luminescence Properties of KMnF_3_: Yb, Er NPs

After the introduction of Ce^3+^ ions, the luminescence intensity at 1.53 μm of Er^3+^ ions was significantly increased. In KMnF_3_: Yb, Er, Ce NPs, all the energy transfer processes were shown in Figure 8. The Yb^3+^ ions were excited from the ^2^F_7/2_ to the ^2^F_5/2_ energy level under the excitation of the 980 nm laser diode, and the energy was transferred to the Er^3+^ ions to accumulate the higher energy level of the Er^3+^ ions: ^2^H_9/2_, ^4^F_7/2_, ^2^H_11/2_, ^4^S_3/2_, ^4^F_9/2_. Among them, ^4^H_11/2_ → ^4^I_15/2_ (≈525 nm), ^4^S_3/2_ → ^4^I_15/2_ (≈545 nm), and ^4^F_9/2_ → ^4^I_15/2_ (655 nm) transitions give up-conversion emission. The ^4^I_13/2_ → ^4^I_15/2_ transition gives luminescence at 1.53 μm. Interestingly, the energy transfer occurs between Ce^3+^ ions and Er^3+^ ions with the introduction of Ce^3+^ ions: ^4^I_11/2_ (Er^3+^) + ^2^F_5/2_ (Ce^3+^) → ^4^I_13/2_ (Er^3+^) + ^2^F_7/2_ (Ce^3+^). As a result, the ^4^I_11/2_ energy levels of the Er^3+^ ions were populated at the ^4^I_13/2_ energy level [30,31,32]. The intensity of up-conversion luminescence was reduced, while the luminescence intensity at 1.53 μm was improved after the introduction of Ce^3+^. As shown in the energy transfer schematic in Figure 8. The energy of the excitation laser at 980 nm is first transferred to the Yb^3+^ ions, and then the energy is transferred to the ^4^I_11/2_ energy level of the Er^3+^ ions through the energy transfer between the Yb^3+^ ions and the Er^3+^ ions. The ^4^I_11/2_ energy level of the Er^3+^ ions can continue to receive energy transferred from the Yb^3+^ ions, thereby increasing the population of the other excited state energy levels of the Er^3+^ ions. After the introduction of Ce^3+^ ions, the energy transfer process occurs between Ce^3+^ ions and Er^3+^ ions. The population of the ^4^I_11/2_ energy level of Er^3+^ ions decreases and the population of the ^4^I_13/2_ energy level of Er^3+^ ions increases. The decrease of the population of the ^4^I_11/2_ energy level of Er^3+^ ions leads to the decrease of the population of all other excited state energy levels of Er^3+^ ions and the weakening of the up-conversion luminescence intensity. The increase of the population of the ^4^I_13/2_ energy level of Er^3+^ ions leads to the increase of the luminescence intensity at 1.5 μm (^4^I_13/2_ → ^4^I_15/2_). As shown in the Figure 9, after the introduction of Ce^3+^ ion, the intensity of luminescence of the sample at 1.5 μm was significantly increased, while the up-conversion luminescence intensity of the sample was significantly decreased. With the increasing Ce^3+^ ion concentration, the luminescence intensity of the sample at 1.5 μm has been increased, and the up-conversion luminescence intensity has been decreased. When the Ce^3+^ ion reaches the optimal doping concentration, the luminescence intensity of the sample reaches the maximum at 1.5 μm. As the Ce^3+^ ion concentration continued to increase, the luminescence intensity at 1.5 μm of the sample began to weaken.

In order to investigate the best concentration of Ce^3+^ ions, we measured the up-conversion luminescence spectrum of KMnF_3_:18Yb, 2Er, xCe (x = 1%, 2%, 3%, 4%, 5%) under the excitation of a 980 nm laser diode, the data was shown in Figure 9a. The luminescence peaks were attributed to the ^2^H_11/2_ → ^4^I_15/2_ (525 nm), ^4^S_3/2_ → ^4^I_15/2_ (545 nm), ^4^F_9/2_ → ^4^I_15/2_ (655 nm).

The luminescence at 1.53 μm of the NPs under the excitation of a 980 nm laser diode gradually increased with the increase of Ce^3+^ concentration (as shown in Figure 9b). This is due to the energy transfer between Ce^3+^ and Er^3+^: ^4^I_11/2_ (Er^3+^) + ^2^F_5/2_ (Ce^3+^) → ^4^I_13/2_ (Er^3+^) + ^2^F_7/2_ (Ce^3+^). This results in an increase in the population of the ^4^I_13/2_ level of the Er^3+^ ion and a decrease in the population of the ^4^I_11/2_ level. The increase of the population of the ^4^I_13/2_ energy level greatly increases the luminescence intensity of the Er^3+^ ion at 1.5 μm. When the Ce^3+^ concentration was 4 mmol%, the luminescence intensity at 1.53 μm of KMnF_3_: Yb, Er, Ce NPs reaches the maximum under excitation of the 980 nm laser diode (as shown in Figure 9d). As the Ce^3+^ concentration continues to increase, the luminescence intensity at 1.53 μm begins to weaken, which was due the concentration quenching. Through the above experiments, we determined that the best Ce^3+^ doping concentration in KMnF_3_: Yb, Er NPs was 4%. This Ce^3+^ doping concentration will be used in the subsequent core-shell coating experiments.

#### 3.2.3. Up-Conversion Luminescence and Luminescence at 1.53 μm Characteristics of Core-Shell KMnF_3_: Yb, Er, Ce@KMnF_3_: Yb NPs

We compared the up-conversion luminescence spectrum and luminescence at 1.53 μm of KMnF_3_:18Yb, 2Er, 4Ce core NPs with KMnF_3_:18Yb, 2Er, 4Ce@KMnF_3_:20Yb core-shell NPs under excitation of 980 nm laser diode (as shown in Figure 10). It can be seen that when the active shell layer was coated on the surface of the core NPs, the intensity of up-conversion luminescence and luminescence intensity at 1.53 μm was significantly enhanced. Due to this, the insertion of the shell can inhibit the non-radiative transition [23,24,33], and the Yb^3+^ ions in the shell can transfer energy from the pump source to the core. Thereby, the active shell contributes to the enhancement of the luminescence intensity of the up-conversion and luminescence intensity at 1.53 μm [34].

In addition, the lifetime of the ^4^I_13/2_ energy level of Er^3+^ in KMnF_3_: 18Yb, 2Er, KMnF_3_:18Yb, 2Er, xCe (x = 1%, 2%, 3%, 4%, 5%) and KMnF_3_:18Yb, 2Er, 4%Ce@KMnF_3_: 20Yb NPs was measured under excitation of 980 nm pulsed laser with a pulse width of 400 µs and a frequency of 50 HZ as the excitation source. The result was shown in Figure 11. Each of the delay curves can be fitted well with a single-exponential function as $I=I_{0}\exp\left(-t/\tau \right)$, where $I_{0}$ is the initial emission intensity at t=0 and τ is the lifetime of the monitored level. We found that after the introduction of Ce^3+^ ions, the lifetime of the ^4^I_13/2_ energy level was increased. This was because after the introduction of Ce^3+^ ions, the energy transfer occurs between Ce^3+^ ions and Er^3+^ ions with the introduction of Ce^3+^ ions: ^4^I_11/2_ (Er^3+^) + ^2^F_5/2_ (Ce^3+^) → ^4^I_13/2_ (Er^3+^) + ^2^F_7/2_ (Ce^3+^). After energy transfer occurs, the population of ^4^I_13/2_ level of Er ion increases, which leads to the increase of the lifetime of ^4^I_13/2_ level. The lifetime of the ^4^I_13/2_ level increases with the increase of Ce ion concentration. When the doping concentration of Ce ion reaches 4%, the luminescence of the Er ion at 1.5 μm and the lifetime of the ^4^I_13/2_ energy level reach the maximum. With the continuous increase of Ce ion doping concentration, the luminescence of the Er ion at 1.5 μm and the lifetime of ^4^I_13/2_ level will decrease, which is caused by concentration quenching. In a series of lifetime curves, the core-shell structure sample has the longest lifetime. This is due to growing an active shell on the core NPs, and the increase in lifetime is due to the surface passivation effect, which leads to a reduction in nonradiative relaxation rate.

## 4. Conclusions

In summary, we prepared KMnF_3_ NPs by a hydrothermal method. Through multiple sets of control experiments, we found the best concentration of Yb^3+^ and Er^3+^ when the KMnF_3_: Yb, Er has the strongest luminescence at 1.53 μm under excitation of 980 nm laser diode. The introduction of Ce^3+^ ions enhanced the luminescence intensity at 1.53 μm of KMnF_3_: Yb, Er NPs, and the optimal concentration of Ce^3+^ was found. The preparation of the core-shell structure further enhanced luminescence intensity at 1.53 μm of KMnF_3_: Yb, Er NPs. It can be seen from the spectrum that after the active shell was coated, the intensity of up-conversion luminescence and luminescence intensity at 1.53 μm of the NPs under the excitation of 980 nm laser diode was greatly enhanced; this also shows that the core-shell structure can well inhibit the surface quenching effect. Since KMnF_3_: Yb, Er, Ce@KMnF_3_: Yb NPs have good broadband luminescence properties at 1530 nm, they can be the potential gain medium in the future polymer-based waveguide amplifiers. 

## Figures and Tables

**Figure 1 nanomaterials-13-02153-f001:**
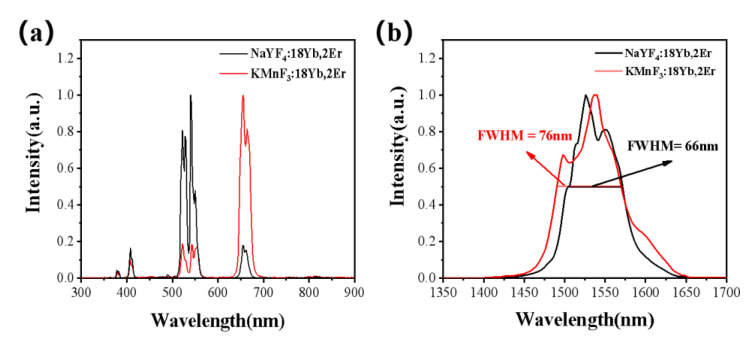
The normalized comparison of the (**a**) up-conversion spectrum and (**b**) emission spectrum at 1.53 μm of KMnF_3_: 18Yb, 2Er and NaYF_4_: 18Yb, 2Er under the excitation of a 980 nm laser diode.

**Figure 2 nanomaterials-13-02153-f002:**
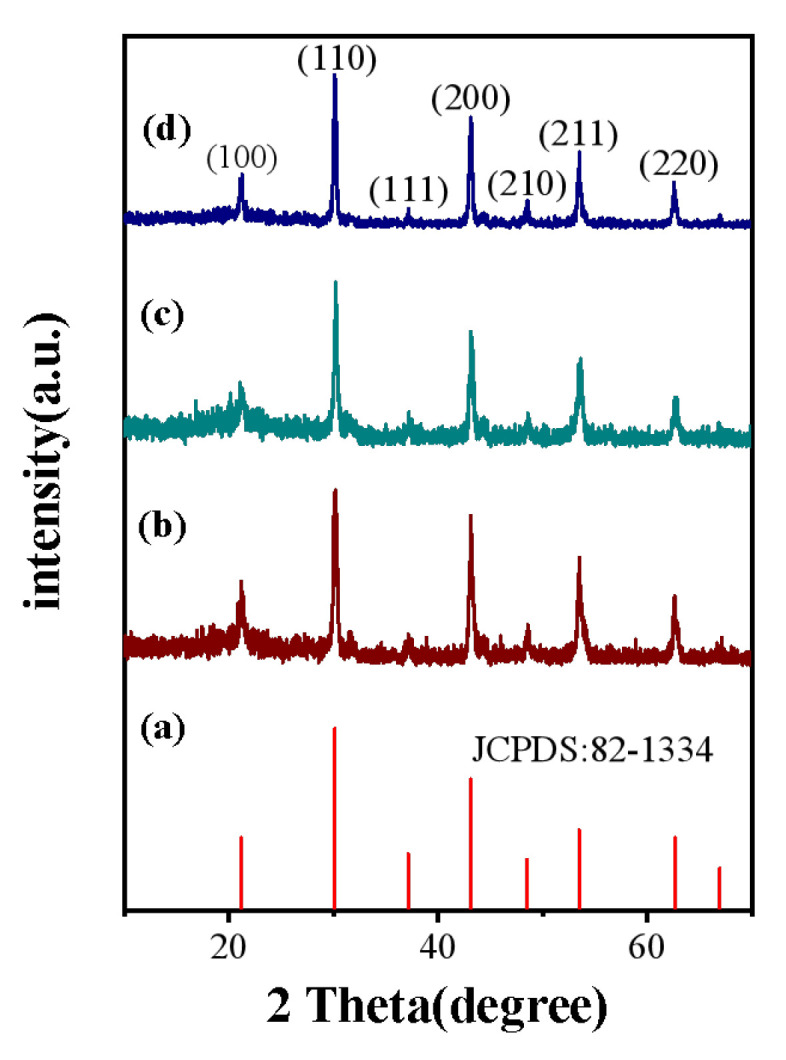
XRD patterns of (**a**) KMnF_3_ standard card; (**b**), KMnF_3_: 18Yb, 2Er; (**c**), KMnF_3_: 18Yb, 2Er, 4Ce; (**d**), KMnF_3_: 18Yb, 2Er, 4Ce@KMnF_3_: 20Yb.

**Figure 3 nanomaterials-13-02153-f003:**
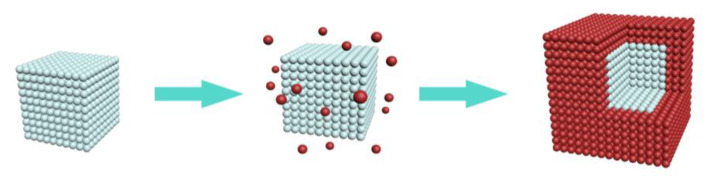
Schematic diagram of KMnF_3_: 18Yb, 2Er, 4Ce core NPs and KMnF_3_: 18Yb, 2Er, 4Ce@KMnF_3_: 20Yb core-shell NPs.

**Figure 4 nanomaterials-13-02153-f004:**
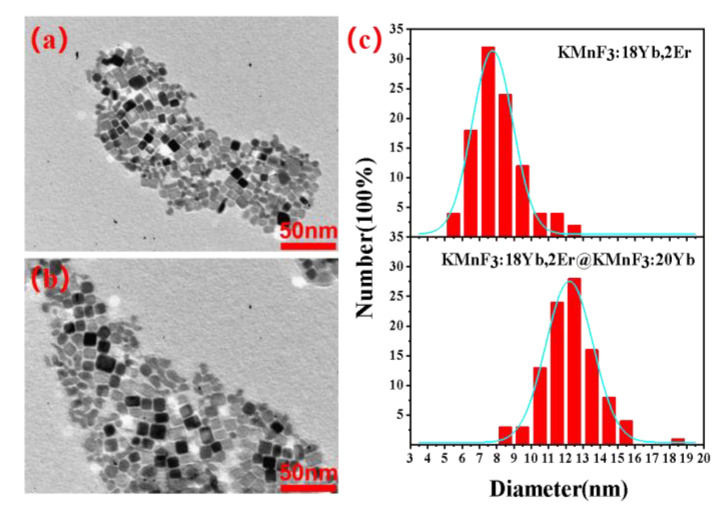
(**a**) TEM of the KMnF_3_: 18Yb, 2Er, 4Ce core NPs; (**b**) TEM of the KMnF_3_: 18Yb, 2Er, 4Ce@KMnF_3_: 20Yb core-shell structured NPs; (**c**) KMnF_3_:18Yb, 2Er, 4Ce and KMnF_3_: 18Yb, 2Er, 4Ce@KMnF_3_: 20Yb particle size distribution.

**Figure 5 nanomaterials-13-02153-f005:**
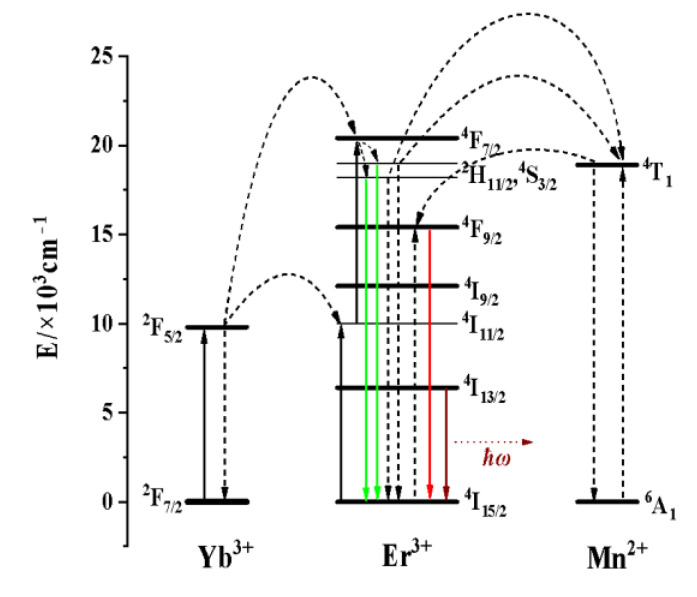
Schematic diagram of energy transfer in KMnF_3_: Yb, Er nanoparticles under the excitation of a 980 nm laser diode.

**Figure 6 nanomaterials-13-02153-f006:**
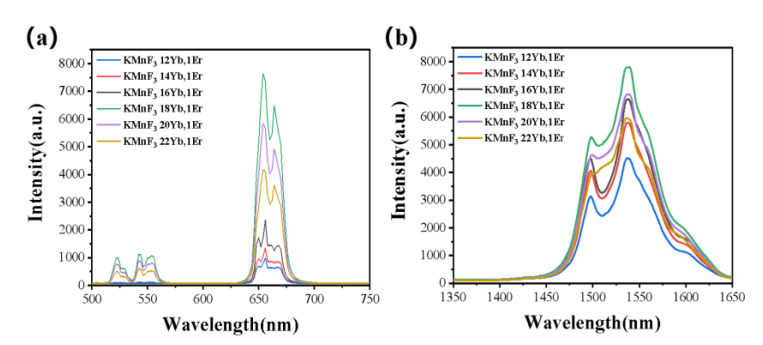
The (**a**) up-conversion emission spectrum and (**b**) emission spectrum at 1.53 μm of KMnF_3_: xYb, 1Er (x = 12%, 14%, 16%, 18%, 20%, 22%) NPs excited by a 980 nm laser diode.

**Figure 7 nanomaterials-13-02153-f007:**
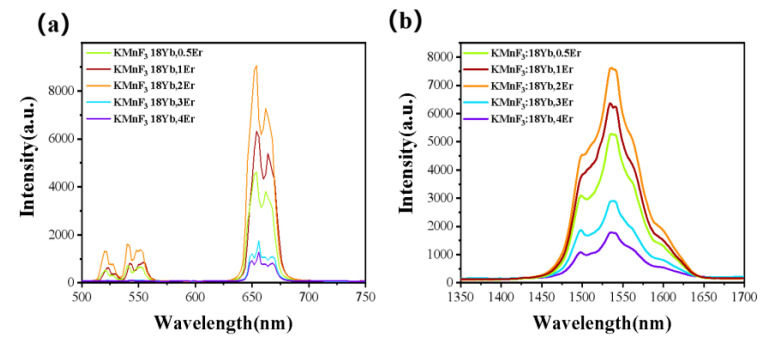
The (**a**) up-conversion emission spectrum and (**b**) emission spectrum at 1.53 μm of KMnF_3_: 18Yb, xEr (x = 0. 5%, 1%, 2%, 3%, 4%) NPs excited by a 980 nm laser diode.

**Figure 8 nanomaterials-13-02153-f008:**
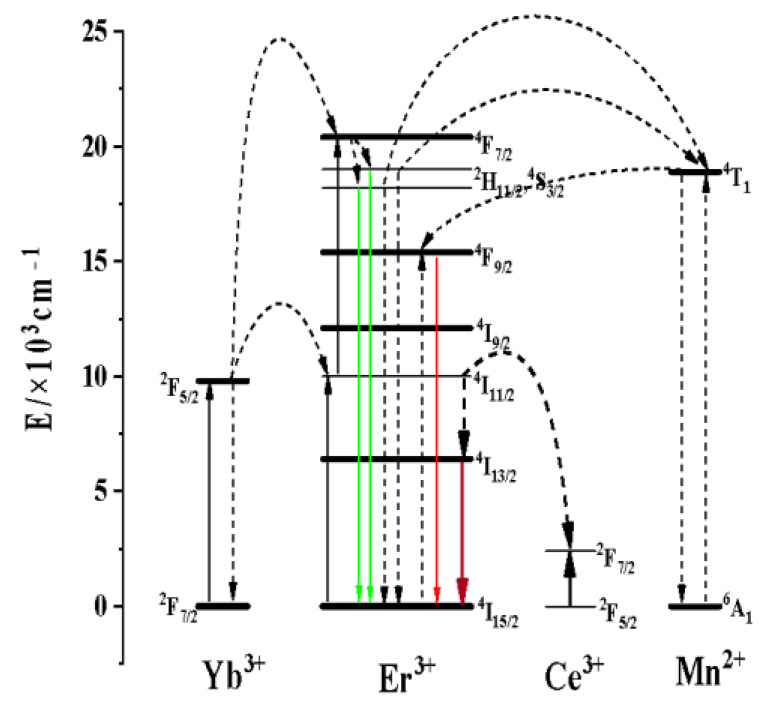
Schematic diagram of energy transfer process in KMnF_3_: Yb, Er, Ce NPs.

**Figure 9 nanomaterials-13-02153-f009:**
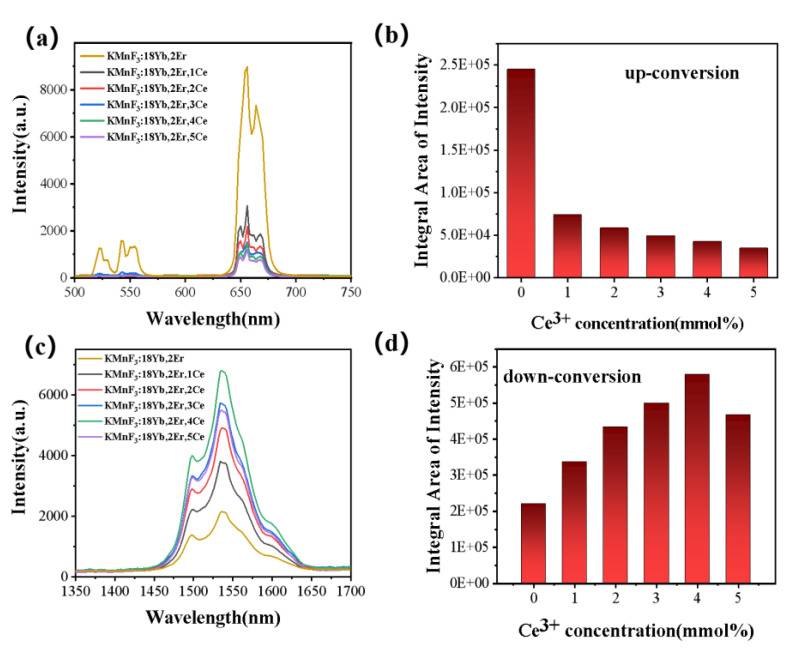
KMnF_3_: 18Yb, 2Er, xCe (x = 1%, 2%, 3%, 4%, 5%) under the excitation of 980 nm laser diode (**a**) up-conversion luminescence spectrum and (**b**) luminescence spectrum at 1.53 μm and (**c**) up-conversion luminescence spectrum and (**d**) luminescence spectrum at 1.53 μm integral area under different Ce^3+^ concentrations.

**Figure 10 nanomaterials-13-02153-f010:**
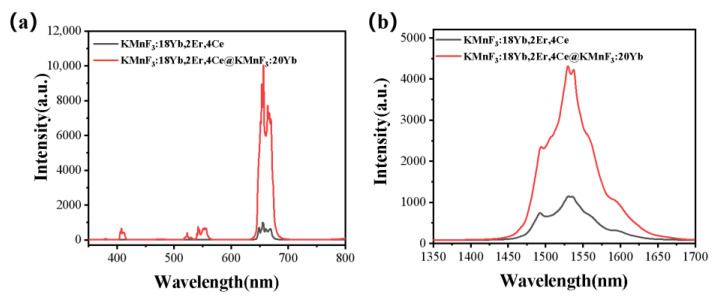
(**a**) Up-conversion luminescence spectrum and (**b**) luminescence spectrum at 1.53 μm of KMnF_3_: 18%Yb, 2%Er, 4%Ce and KMnF_3_: 18%Yb, 2%Er, 4%Ce@KMnF_3_: 20%Yb excited by a 980 nm laser diode.

**Figure 11 nanomaterials-13-02153-f011:**
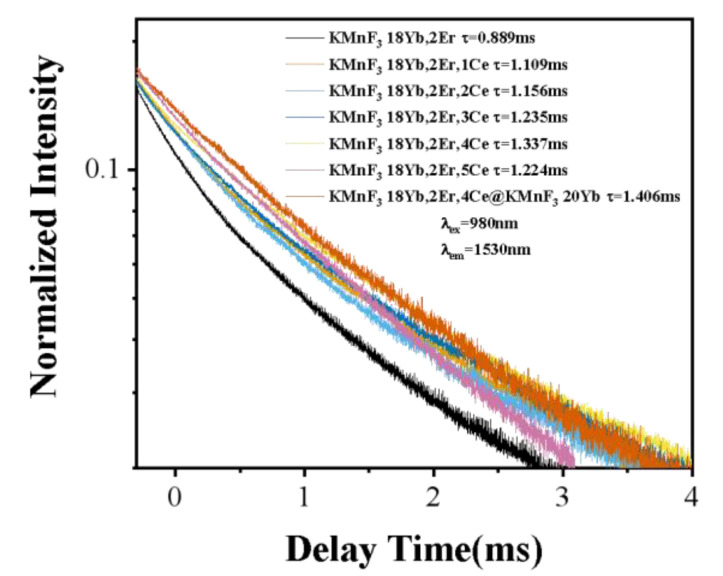
Photoluminescence decay curves of the ^4^I_13/2_ level of Er^3+^.

## Data Availability

The data supporting the findings of this study are available from the authors upon reasonable and appropriate request.

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
