# Peer review of "Introduce Ce3+ Ions to Realize Enhancement of C+L Band Luminescence of KMnF3: Yb, Er Nanoparticles"

_nanomaterials, 2023, doi:10.3390/nano13152153_

Round 1

Reviewer 1 Report

The manuscript is aimed to study upconversion and downconversion processes in fluoride crystals doped with Er, Yb. The authors found some interesting effects, however the novelty and explanation is not fully clear. There are the following major comments:

1) The given mechanism involving Ce ions is not clear (Fig. 8) and it is not sufficiently proved by experimental data.

2) The important role plays charge compensation mechanism when co-doping 3+ lanthanide occurs. The possible point groups of Er3+, Ce3+ and Yb3+ ions should be discussed from analysis of luminescence band structure of upconversion and photoluminescence band under direct f-f excitation.

3) The Er3+ ions need not Yb. The upconversion of Er-Er is also registered. Authors should compare the deposit of Er-Er and Yb-Er upconversion processes. For example, the similar analysis has been provided in [10.1039/C9TC06591A] and [10.1134/S0030400X20110211]

4) The downconversion is incorrect term in this manuscript. Downconversion means multiphoton emission process under excitation by one photon. This should be proved by absolute quantum yield measurements. The observed luminescence in the manuscript should be explained using simple energy transfer and relaxation between lanthanides due to their high concentration.

Reviewer 2 Report

The study investigated the metal ion-assisted nanoparticle structures which are synthesized via the hydrothermal method. This work is an “exciting issue” and recent developments in this field have led to a surge of interest from the research community. Generally, this study is well-written, logically, and coherently in literature. The authors showed they are experts in this field. Therefore, it deserves to be published in Nanomaterials after a minor revision as follows:

The author should add new references associated with the conventional metal nanoparticle materials to obviously point out the main flow and the critical point of this project as follows:

Add new refs [1-3] in Introduction Part for the conventional metal nanoparticles at: “Rare earth ion-doped nanoparticles (NPs) have attracted much attention since their excellent optical properties compared with the conventional metal nanoparticles [1-3].”

1. Introduction to metallic nanoparticles. Journal of Pharmacy and Bioallied Sciences 2(4):p 282-289.

2. Biosynthesis of silver nanoparticles for biomedical applications: A mini review. Inorganic Chemistry Communications 2022, 145, 109980

3. Biosynthesis and Characterizations of Silver Nanoparticles from Annona squamosa Leaf and Fruit Extracts for Size-Dependent Biomedical Applications. Nanomaterials 2022, 12(4), 616.

Round 2

Reviewer 1 Report

accept

Author Response

Thank you very much for your suggestions. The English writing has been modified as needed, and the issues in the article have been corrected. The references have also been revised as requested.
